# Voltage-based magnetization switching and reading in magnetoelectric spin-orbit nanodevices

Diogo C. Vaz [1] ✉, Chia-Ching Lin[2], John J. Plombon[2], Won Young Choi [1,7], Inge Groen[1], Isabel C. Arango [1], Andrey Chuvilin [1,3], Luis E. Hueso [1,3], Dmitri E. Nikonov [2], Hai Li [2], Punyashloka Debashis[2], Scott B. Clendenning[2], Tanay A. Gosavi[2], Yen-Lin Huang[4], Bhagwati Prasad[5], Ramamoorthy Ramesh [4,8], Aymeric Vecchiola [6], Manuel Bibes [6], Karim Bouzehouane [6], Stephane Fusil[6], Vincent Garcia [6], Ian A. Young [2] & Fèlix Casanova [1,3] ✉

As CMOS technologies face challenges in dimensional and voltage scaling, the demand for novel logic devices has never been greater, with spin-based devices offering scaling potential, at the cost of significantly high switching energies. Alternatively, magnetoelectric materials are predicted to enable low-power magnetization control, a solution with limited device-level results. Here, we demonstrate voltage-based magnetization switching and reading in nanodevices at room temperature, enabled by exchange coupling between multiferroic $BiFeO_3$ and ferromagnetic CoFe, for writing, and spin-to-charge current conversion between CoFe and Pt, for reading. We show that, upon the electrical switching of the $BiFeO_3$, the magnetization of the CoFe can be reversed, giving rise to different voltage outputs. Through additional microscopy techniques, magnetization reversal is linked with the polarization state and antiferromagnetic cycloid propagation direction in the $BiFeO_3$. This study constitutes the building block for magnetoelectric spin-orbit logic, opening a new avenue for low-power beyond-CMOS technologies.

After 50 years of continuous transistor size downscaling and increased performance[1], future iterations of logic circuits will require beyond-CMOS alternatives[2] that explore new physical effects through non-conventional materials. While Moore's law is still sustained by increasingly complex transistor designs and lithography advances[3], the last decade saw a clear breakdown of Dennard's scaling, where smaller transistors no longer mean smaller operational voltages, compromising the energy efficiency and performance of future chips. In recent years, a flurry of new logic devices has emerged, driven by the usage of alternative state variables, such as spin, polarization, strain,

and orbital[4]. Among these options, spin-based solutions have shown tremendous promise and applicability[5]. Owing to their non-volatile nature, effects like spin-transfer torque (STT) and spin–orbit torque (SOT) brought major improvements in stand-by power, as well as in terms of endurance, writing speed, and compatibility with back-end of line (BEOL) fabrication processes[6,7]. Yet, controlling magnetization states using these methods still requires rather large currents, preventing their usage as a realistic non-volatile logic solution. Alternatively, voltage-based methods gained some traction in recent years[8], mainly pushed by voltage-controlled magnetic anisotropy (VCMA)[9],

[1]CIC nanoGUNE BRTA, 20018 Donostia-San Sebastian, Basque Country, Spain. [2]Components Research, Intel Corp., Hillsboro, OR 97124, USA. [3]IKERBASQUE, Basque Foundation for Science, 48009 Bilbao, Basque Country, Spain. [4]Department of Physics, University of California, Berkeley, CA 94720, USA. [5]Materials Engineering Department, Indian Institute of Science, Bengaluru 560012 Karnataka, India. [6]Laboratoire Albert Fert, CNRS, Thales, Université Paris-Saclay, 91767 Palaiseau, France. [7]Present address: VanaM Inc., 21-1 Doshin-ro 4-gil, Yeongdeungpo-gu, Seoul, Republic of Korea. [8]Present address: Department of Physics and Astronomy, Rice University, Houston, TX 77005, USA. ✉e-mail: diogocastrovaz@gmail.com; f.casanova@nanogune.eu

where voltage-induced dynamic switching of magnetization has been reported[10]. While field-free VCMA writing has been recently shown[11], further work is required to improve the VCMA coefficient, in order to bring this technology closer to product applications.

A pathway for field-free voltage-based switching of magnetism has been proposed using magnetoelectric multiferroics[12]. Among several possible combinations, the coexistence of ferroelectricity and ferromagnetism is expected to allow the control of magnetization through switching of the ferroelectric polarization with an electric field. In this category, $BiFeO_3$ has been the most studied material, exhibiting a tight coupling between antiferromagnetic (AF) and ferroelectric (FE) orders at room temperature. One of the most notable results toward multiferroic-based devices was the demonstration of magnetization reversal by 180° in a CoFe element, exchange coupled with $BiFeO_3$, upon application of an electric field[13]. The result was interpreted considering weak ferromagnetism arising from canting of the $Fe^{3+}$ magnetic moments in $BiFeO_3$[14], which can couple to the magnetization of the CoFe. Upon a two-step switching of the polarization and canted magnetization in $BiFeO_3$, the magnetization of the CoFe is expected to follow this motion and reverse[15].

Since then, the road to multiferroic-based devices has been long and tortuous, with sparse results reported[16]. Yet, it is expected that such devices can bring magnetization writing energies down to the aJ range[17], an improvement of several orders of magnitude when compared with state-of-the-art current-based devices. This driving force led to the recent proposal of magnetoelectric spin–orbit (MESO) logic[17], suggesting a spin-based nanodevice adjacent to a multiferroic, where the magnetization is switched solely with a voltage pulse and is electrically read using spin-to-charge current conversion (SCC) phenomena.

In this article, we demonstrate the experimental implementation of such a device. We fabricate SCC nanodevices on $BiFeO_3$ and analyze the reversibility of the magnetization of CoFe using a combination of piezoresponse (PFM) and magnetic force microscopy (MFM), where the polarization state of the $BiFeO_3$ and the magnetization of CoFe are imaged upon switching. We then correlate this with all-electrical SCC experiments where voltage pulses are applied to switch the $BiFeO_3$, reversing the magnetization of CoFe (writing) and different SCC output voltages are measured depending on the magnetization direction (reading). Lastly, we investigate the magnetic textures at the surface of $BiFeO_3$ using scanning nitrogen-vacancy (N–V) magnetometry, where the coupling between CoFe and $BiFeO_3$ is linked with the AF cycloid propagation direction.

## Results

In Fig. 1a, we show a sketch of the fabricated MESO nanodevice. The MESO concept can be described as an assembly of a magnetoelectric (ME) module used for writing and a spin–orbit (SO) module used for reading[18]. The ME module comprises the multiferroic and an adjacent ferromagnet, here $BiFeO_3$ and CoFe, respectively. Voltages pulses ($V_p$) are applied between a metallic $La_{0.7}Sr_{0.3}MnO_3/SrRuO_3$ bottom electrode and the CoFe, so that the polarization ($P$) and the AF order ($L$) in the $BiFeO_3$ can be switched, as exemplified in Fig. 1b. Here, the magnetization direction of CoFe ($M_{CoFe}$) is also reversed, following the reversal of $P$ and $L$, due to exchange coupling at the CoFe/$BiFeO_3$ interface. The SO module is based on a T-shaped nanostructured device composed of CoFe and the SO material Pt, following a recent study on SCC for magnetic state readout[19]. A spin-polarized current ($I_{in}$) is electrically driven from CoFe to Pt, where, at the Pt/CoFe junction, the spins are converted into a charge current through the inverse spin Hall effect (ISHE) and picked up as a transverse voltage $V_{SO}$. Depending on the magnetization direction, spins $\sigma$ are deflected either to the right or left, as shown in Fig. 1b, enabling a fully electrical method of magnetization state readout, that, in addition, generates an electromotive force that can drive another circuit element. The device is based on a Pt(10 nm)/CoFe(2.5 nm)/$BiFeO_3$(30 nm)/$La_{0.7}Sr_{0.3}MnO_3$(4 nm)/$SrRuO_3$(10 nm) stack grown on a $DyScO_3$ (110) substrate, using a combination of pulsed laser deposition and sputtering (see details in Methods). The fabrication of the device comprises positive nanolithography processes using e-beam lithography, Ar-ion

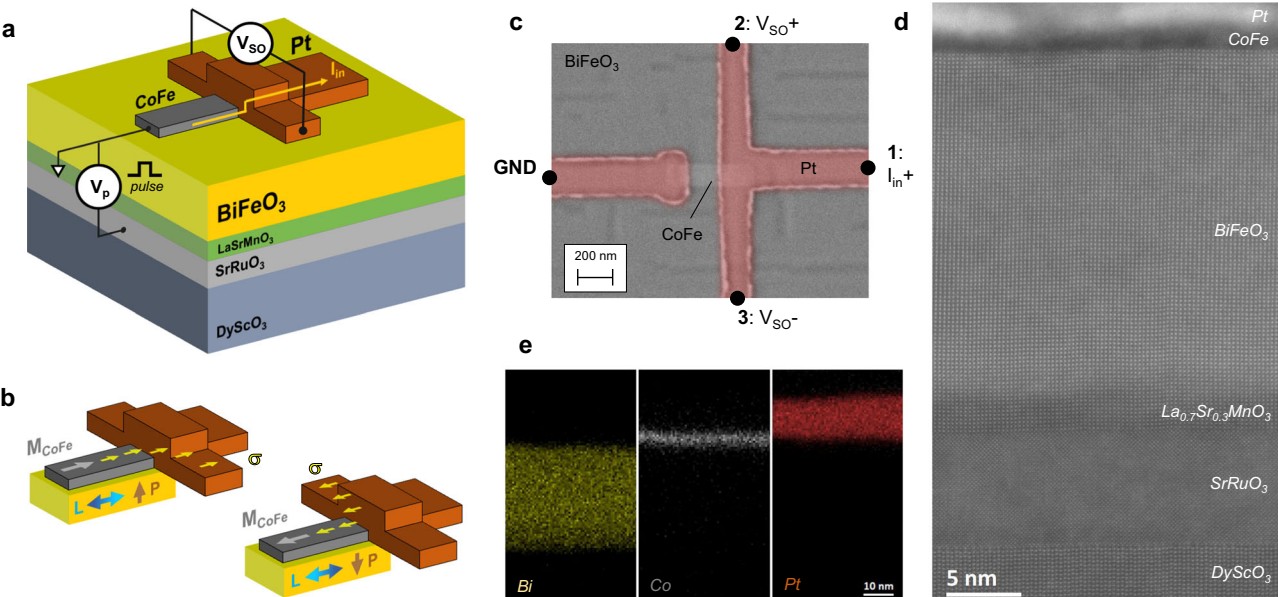

**Fig. 1 | MESO nanodevice and material characterization. a** MESO device configuration composed of a $DyScO_3$ substrate, $La_{0.7}Sr_{0.3}MnO_3/SrRuO_3$ bottom electrodes, multiferroic $BiFeO_3$, magnetic CoFe element and SO material Pt. The logic state variable is given by the magnetization direction in CoFe. **b** Writing is achieved by applying voltage pulses $V_p$ between the CoFe and the bottom electrode, switching the polarization $P$ and AF order $L$ of $BiFeO_3$, which reverses the magnetization $M_{CoFe}$ of CoFe. Reading is achieved through ISHE, where a spin-polarized current $I_{in}$ is injected into Pt, leading to different transverse output voltages $V_{SO}$, depending on the initial orientation of the injected spins $\sigma$. **c** SEM top-view image of the fabricated nanodevice. $I_{in}$ is applied between lead 1 and ground GND, and $V_{SO}$ is detected between leads 2 and 3. **d** TEM cross-sectional image at the Pt/CoFe junction region on the fabricated nanodevice. **e** EDX elemental maps of Bi (from the $BiFeO_3$ layer), Co (from the CoFe layer) and Pt at the Pt/CoFe junction region.

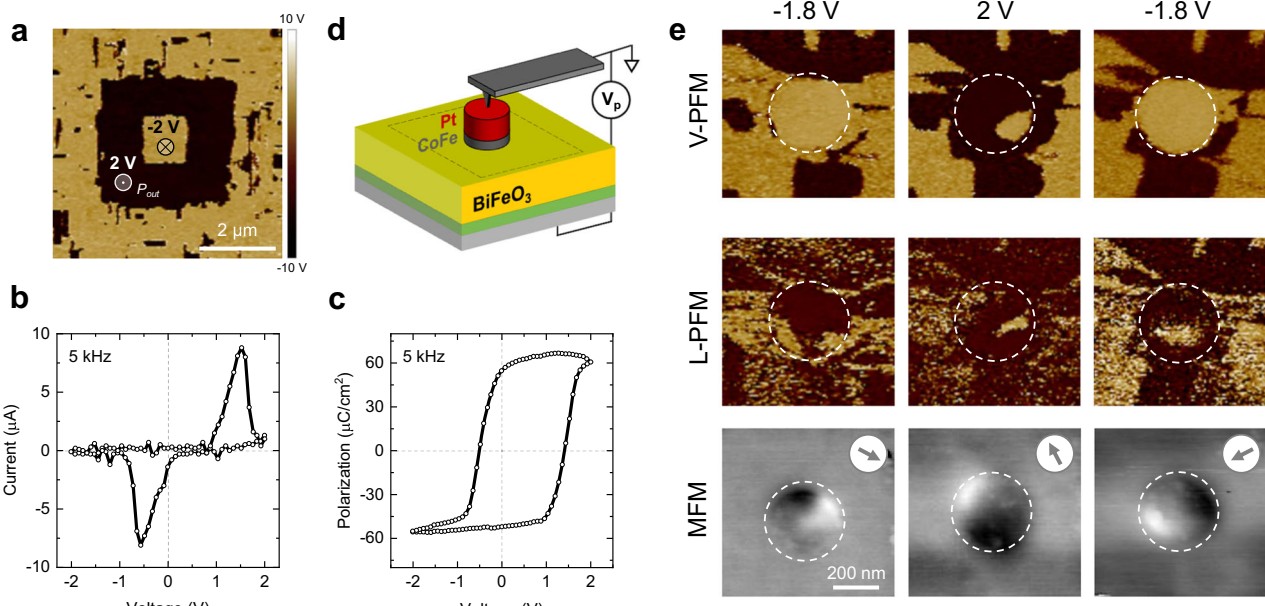

**Fig. 2 | PFM and MFM characterizations. a** Out-of-plane polarization $P_{out}$ after a box-in-box switching experiment on the bare BiFeO$_3$ surface. Dark and bright areas correspond to polarization poled up and down, respectively. **b** Current and **c** polarization vs. voltage loops on Pt/CoFe disks 5 μm in diameter over BiFeO$_3$/SrRuO$_3$/DyScO$_3$, collected with a frequency of 5 kHz. **d** Sketch of the PFM and MFM experiments. Dashed line corresponds to the area scanned with PFM and MFM.

**e** Out-of-plane (V-) and in-plane (L-) PFM phase images after applying voltage pulses of −1.8 V, 2 V, and −1.8 V to a Pt/CoFe disk 300 nm in diameter, showing the FE domains in BiFeO$_3$ underneath the disk. Corresponding MFM images showing the magnetization direction of the CoFe after each pulse, represented by the gray arrows.

milling, and sputtering, used to define both the CoFe wire (500 nm × 150 nm in lateral size), the Pt T-shaped electrode, and contacts. The device is capped with SiO$_2$(5 nm) to prevent oxidation of the CoFe. Details of the fabrication process flow can be found in Supplementary Information Note 1. A scanning electron microscopy (SEM) top image of the integrated MESO nanodevice is shown in Fig. 1c, and a cross-sectional image of the Pt/CoFe junction area, taken by transmission electron microscopy (TEM) after device fabrication, is shown in Fig. 1d. We observe highly ordered epitaxial growth of the oxide hetero-structure, as well as clean and sharp interfaces between BiFeO$_3$, CoFe, and Pt. From the energy dispersive X-ray spectroscopy (EDX) maps shown in Fig. 1e, we observe minimal interdiffusion between the three layers.

We start by investigating the magnetization orientation of CoFe upon polarization reversal of the BiFeO$_3$ using a combination of PFM and MFM. In Fig. 2a, we observe that the polarization of the bare BiFeO$_3$ can be poled up (dark area) and down (bright area) with posi-tive (2 V) and negative (−2 V) voltages, respectively. Then, a CoFe/Pt disk 5 μm in diameter (patterned similarly to the MESO devices) was used to measure the current and polarization vs. voltage loops, as shown in Fig. 2b, c, respectively. These capacitors based on a 30-nm-thick BiFeO$_3$ show large saturation polarization (close to the bulk value), with low leakage, as well as low switching voltages. Indeed, we observe switching voltages of −0.5 V and 1.5 V underneath the metallic disk. A relatively large imprint, normally observed in thin ferroelectric films due to top and bottom contact electrostatic asymmetries, is still present, even though largely improved by the La$_{0.7}$Sr$_{0.3}$MnO$_3$ buffer layer[20]. As illustrated in Fig. 2d, disks 300 nm in diameter were then used to evaluate both the out-of-plane ($P_{out}$) and in-plane ($P_{in}$) polar-ization direction in the BiFeO$_3$ underneath the disk, as well as the direction of $M_{CoFe}$, labeled as V-PFM, L-PFM, and MFM in Fig. 2e, respectively (setup details in Methods).

From V-PFM data, we observe that $P_{out}$ can be reversed between down and up states, after applying voltage pulses of −1.8 V and 2 V, respectively. Small unswitched patches were occasionally observed,

either caused by incomplete switching after $V_p$ is applied or by relaxation back to the $P_{out}$ down state after the voltage pulse is applied (i.e., while $V = 0$ V) due to the BiFeO$_3$ imprint. While $P_{out}$ reverses consistently, we observe from L-PFM data that $P_{in}$ remains split into randomly distributed FE domains for the three voltage pulses applied. For 9 different devices probed in the same sample, $P_{out}$ is always switched, while $P_{in}$ exhibits mostly slight changes in the FE domain structure, suggesting a combination of local 71°/109°/180° switch of the polarization[13]. From MFM, we observe that, after poling the BiFeO$_3$ down with $V_p = −1.8$ V, $M_{CoFe}$ points diagonally to the bottom right. Poling the BiFeO$_3$ up with $V_p = 2$ V reverses the magnetization of CoFe by nearly 180°. Poling the polarization back down with $V_p = −1.8$ V drives a rotation of $M_{CoFe}$ by -90°. Out of the 9 devices tested, the magnetization could always be switched in 3 of them (33.5%), could be partially/randomly switched in 4 (44.5%), and could never be switched in 2 (22%). Out of 24 out-of-plane polarization switching events, we observed that the magnetization switched 13 times (54%) and did not switch 11 times (46%). Extended data and additional switching experiments can be found in Supplementary Information Note 2.

Given these results, we conclude that the reversal of $M_{CoFe}$ is still possible even though there is a lack of control of $P_{in}$, which should be intimately related to the in-plane component of the AF order and canted magnetization in BiFeO$_3$. Moreover, $P_{out}$ switching seems to be driving the reversal/rotation of $M_{CoFe}$, but does not always guar-antee it, indicating that the magnetic configuration in a uniformly out-of-plane polarized region may be more complex, a hypothesis investigated further ahead in this article. Regardless, these results reveal that the magnetization can be manipulated in nanoscale magnets interfaced with BiFeO$_3$ using only a voltage pulse and without external magnetic fields, experimentally demonstrating the MESO writing capabilities.

We now move to the electrical experiments on the MESO nano-devices. First, we investigate the switching ability of the BiFeO$_3$ by applying voltage pulses with a duration of 200 μs between the CoFe element and the bottom of the BiFeO$_3$. After each pulse, we perform

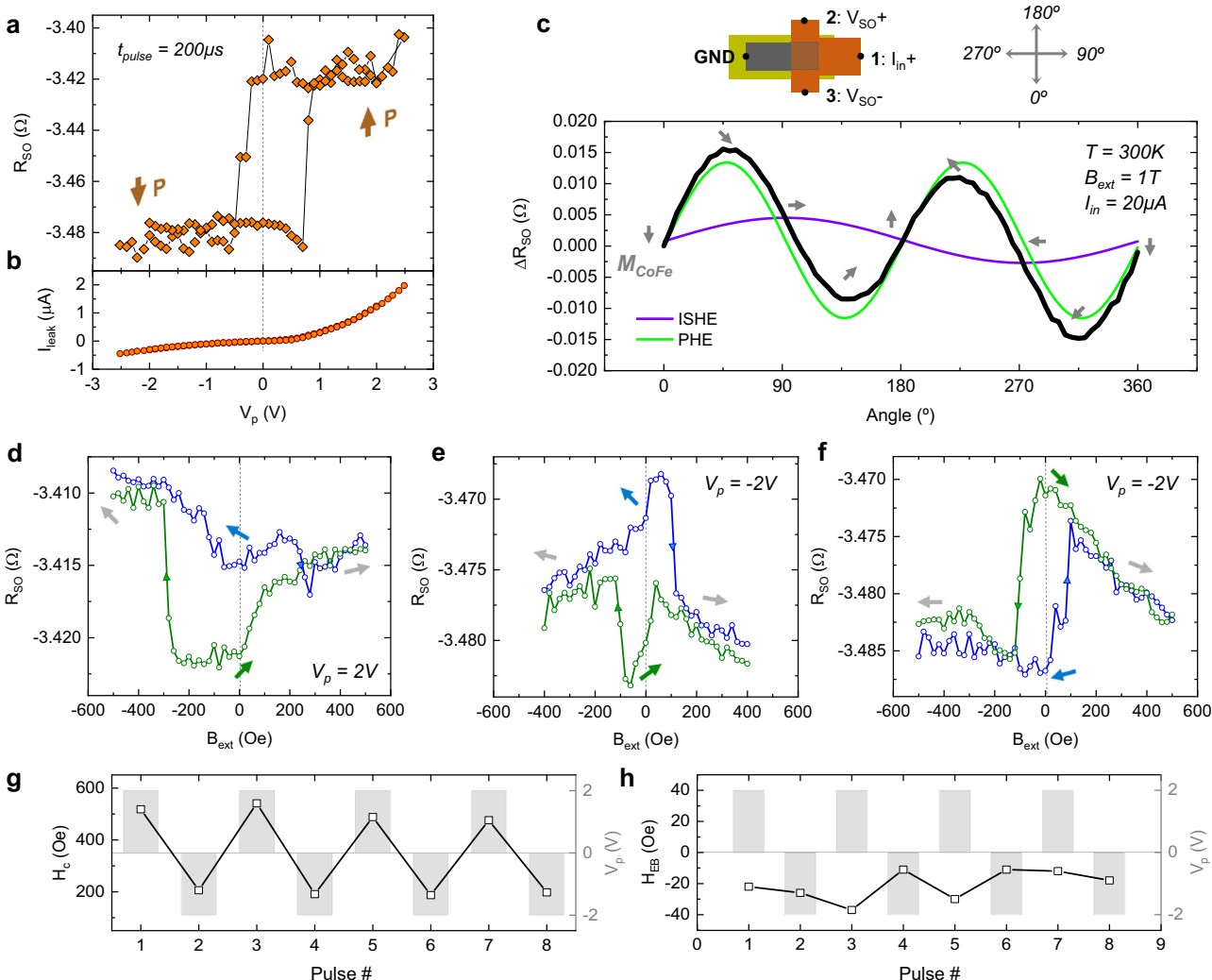

**Fig. 3 | Electrical characterization of MESO nanodevices. a** Baseline of the output resistance $R_{SO}$ and **b** leakage current $I_{leak}$ as a function of the voltage pulse $V_p$ applied between the Pt/CoFe nanodevice and the back of the BiFeO$_3$. Two resistance states are visible depending on the polarization $P$. Resistances are collected 1 s after the pulse is applied. **c** Reading voltage $\Delta R_{SO} = R_{SO} - R^0_{SO}$ as a function of the angle of an in-plane external magnetic field $B_{ext} = 1\,T$ (black curve), after $V_p = -2\,V$, where $R^0_{SO}$ represents the resistance measured at 0°. The data is decomposed in an inverse spin Hall effect (ISHE, in violet) and a planar Hall effect (PHE, in green)

component. Gray arrows represent the magnetization of CoFe as seen from the top of the device (see top-view sketch above the plot). **d** $R_{SO}$ as a function of $B_{ext}$ applied along the long axis of CoFe, after $V_p = 2\,V$, **e** $V_p = -2\,V$ (inhomogeneous coupling), and **f** $V_p = -2\,V$ (fully reversed). The blue and green curves correspond to a $B_{ext}$ sweep from −500 Oe to 500 Oe and back, respectively. Arrows represent $M_{CoFe}$ as seen from the top. **g** Coercivity $H_c$ and **h** exchange bias $H_{EB}$ of the CoFe element as a function of different voltage pulses alternating between $V_p = 2\,V$ and $V_p = -2\,V$ (gray bars).

SCC experiments on the nanodevice, as illustrated in Fig. 1c, by applying $I_{in} = 20\,\mu A$ and measuring the output voltage $V_{SO}$, hereinafter shown as a resistance $R_{SO} = V_{SO}/I_{in}$. As shown in Fig. 3a, upon switching the BiFeO$_3$, the baseline of the SCC signal acquires two stable states, −3.48 Ω and −3.42 Ω. This baseline resistance reflects the slight misalignment of the CoFe element with respect to the Pt T-shaped electrode, giving rise to either a positive or negative transverse voltage[19]. The shift in baseline resistance can be explained by slight modulation of the resistivity of CoFe, either due to a static field effect from the remanent polarization in the BiFeO$_3$ (Ref. 21), or strain induced by different ferroelastic domains. While this resistance vs. voltage loop does not give quantitative information about the polarization, we observe that the BiFeO$_3$ directly underneath the nanodevices switches at −350 mV and 750 mV, in fair agreement with the results from PFM. As shown in Fig. 3b, the leakage current measured through the BiFeO$_3$ layer during the voltage pulses was minimized to about 0.5–1 μA (for $V_p = \pm2\,V$), largely due to the reduced fabricated area of CoFe and Pt in direct contact with BiFeO$_3$.

To fully characterize the SCC results with respect to the expected $M_{CoFe}$ orientation, we measure $\Delta R_{SO}$ as a function of a rotating in-plane external magnetic field $B_{ext} = 1\,T$, enough to fully saturate the CoFe element (Fig. 3c). Here, $\Delta R_{SO}$ represents the relative variation of $R_{SO}$ as the magnetization rotates, allowing a correspondence with the $R_{SO}$ changes observed in Fig. 3d–f. Gray arrows indicate the $M_{CoFe}$ orientation as seen from above. $R_{SO}$ is dominated by the ISHE at 90° and 270° (in violet) when $M_{CoFe}$ points along the easy axis of the CoFe element, following a $\sin(\theta)$, and by the planar Hall effect (PHE) at diagonal orientations (45°, 135°, 225°, and 315°) (in green), following a $\sin(2\theta)$. Similar behavior was identified in Ref. 22. In Fig. 3d–f, we show $R_{SO}$ as a function of $B_{ext}$ along the CoFe long (easy) axis (90°), after voltage pulses of $V_p = \pm2\,V$. Voltage pulses are applied without any external magnetic field, and the loops are taken by sweeping $B_{ext}$ from −500 Oe to 500 Oe (in blue) and back (in green). These loops only provide information on the inherent interactions between $M_{CoFe}$ and BiFeO$_3$, while the $M_{CoFe}$ direction manipulation right after different $V_p$ is investigated further ahead. After applying $V_p = 2\,V$, the $B_{ext}$ sweep

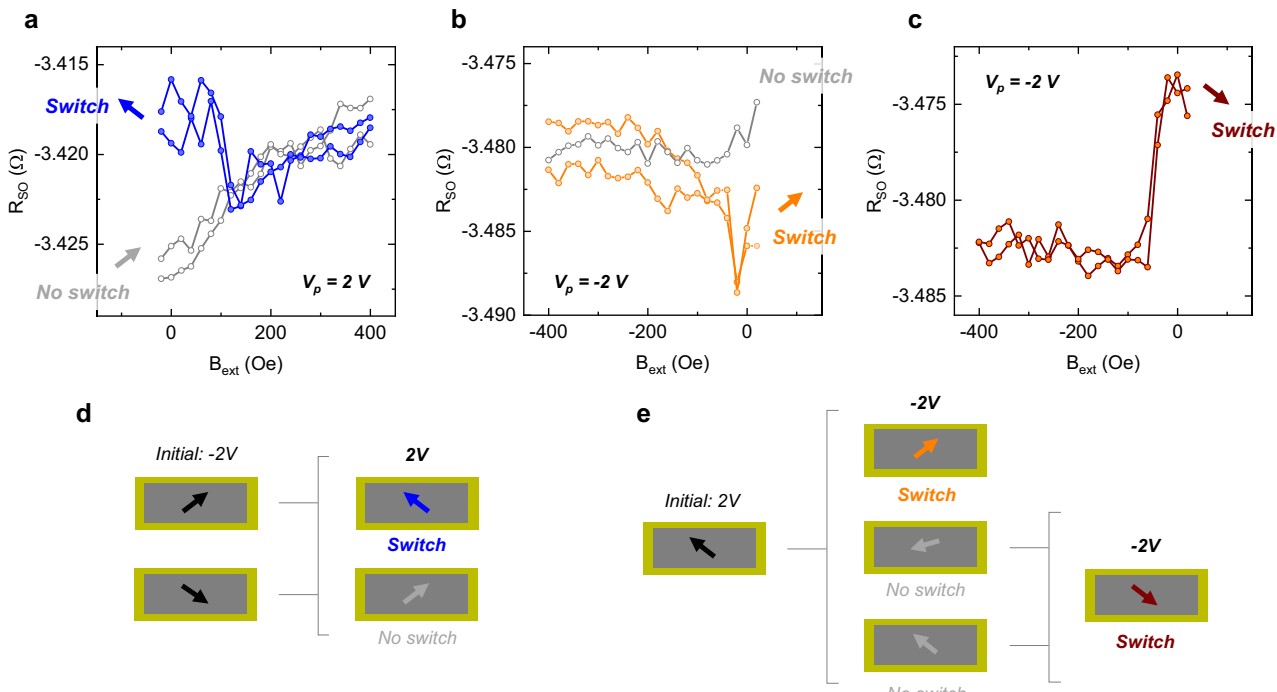

**Fig. 4 | Voltage-based magnetization switching and reading in MESO nanodevices. a** $R_{SO}$ as a function of $B_{ext}$, swept from 0 Oe to 400 Oe, after applying $V_p = 2$ V. The blue curve shows a switch of $M_{CoFe}$ (arrow pointing to the top left), with a higher initial $R_{SO}$ that then reverts to the initial state with increasing $B_{ext}$. The gray curve represents a non-switch event. **b** $R_{SO}$ as a function of $B_{ext}$, swept from 0 Oe to −400 Oe, after applying $V_p = −2$ V. The orange curve shows a switch of $M_{CoFe}$ (arrow pointing to the top right), with a decrease and increase of $R_{SO}$. The gray curve represents a non-switch event. **c** After the field is swept back to 0 Oe, a second pulse $V_p = −2$ V is applied. The red curve shows a switch of $M_{CoFe}$ (arrow pointing to the bottom right), with a higher initial $R_{SO}$ that then reverts to the initial state with increasing negative $B_{ext}$. **d, e** Sketches of every $M_{CoFe}$ switching path after $V_p = 2$ V and $V_p = −2$ V, with the initial magnetization orientation represented by the black arrows, switching events by colored arrows, and non-switching events by gray arrows.

reveals that $R_{SO}$ decreases when $B_{ext}$ approaches zero, indicating that, when mapped to the angle dependence, the magnetization tilts up (Fig. 3d). After applying $V_p = −2$ V, we observe two possible states. In the first case, $M_{CoFe}$ also tilts up around $B_{ext} = 0$ (Fig. 3e). We postulate that, given the occasional incompleteness of the polarization switching observed from PFM, the area underneath the CoFe element may be at times split into different domains, giving rise to inhomogeneous coupling. However, in Fig. 3f the magnetization loop is reversed, and $M_{CoFe}$ tilts down around $B_{ext} = 0$. Additional data concerning the reproducibility of the two fully switched magnetization loops and their correspondence with the angle dependence can be found in Supplementary Information Notes 3 and 4. Unlike T-shaped devices fabricated on Si/SiO$_2$ substrates where at zero external magnetic field $M_{CoFe}$ either points to the right (90°) or to the left (270°) due to shape anisotropy[19], on BiFeO$_3$ the CoFe magnetization may be pulled in any direction, depending on the magnetic textures underneath the CoFe. The observed tilt of $M_{CoFe}$ in our devices suggests that the exchange energy is larger than the shape anisotropy, leading to non-trivial $M_{CoFe}$ orientations in the absence of external magnetic fields. Additionally, we show in Fig. 3g that the CoFe coercivity $H_c$, obtained by the difference between switching fields, changes deterministically between -500 Oe and -200 Oe, as observed in previous reports of exchange coupling at CoFe/BiFeO$_3$ interfaces[23,24]. However, no evident correlation is seen between $V_p$ and the exchange bias $H_{EB}$ (Fig. 3h), obtained by the sum of the switching fields, suggesting prevalent exchange coupling with the AF order[25], rather than the weak ferromagnetism in BiFeO$_3$, that would pull the magnetization in opposite directions depending on the polarization state.

Moving towards a scenario that is closer to the full implementation of MESO logic, we now investigate the $M_{CoFe}$ orientation right after applying $V_p = ±2$ V. We note that, since $M_{CoFe}$ is tilted when $B_{ext} = 0$ T,

the reading function of the MESO device will mostly rely on the PHE instead of the ISHE (see Fig. 3c). While this may reduce the overall output reading voltage, it is still sufficient to electrically probe the magnetization direction in our experiments.

We start by initializing the magnetization direction towards the right by applying a $V_p = −2$ V and sweeping the external magnetic field from 0 to 400 Oe and back. From this state, we apply $V_p = 2$ V at zero magnetic field and measure $R_{SO}$ as a function of $B_{ext}$, to see to which branch of the full loop (Fig. 3d) this half sweep corresponds. As shown in Fig. 4a, a higher initial $R_{SO}$ is observed (in blue), corresponding to a magnetization rotation by either 90° or 180°. Out of eight attempts, this behavior was observed four times (in the same device), with the remaining attempts showing no noticeable change in $R_{SO}$ (in gray).

The magnetization direction was then initialized towards the left by applying $V_p = 2$ V and sweeping the magnetic field from 0 to −400 Oe and back. As shown in Fig. 4b, a first negative voltage pulse $V_p = −2$ V can lead to a magnetization sweep where $R_{SO}$ decreases and then increases (in orange), in close similarity to the lower branch of the full magnetization loop in Fig. 3e, indicating a magnetization rotation of 90°. This behavior was observed three out of nine times, with the remaining sweeps showing no special features (in gray). By bringing the magnetic field back from −400 Oe to 0 Oe, the magnetization is now realigned to the left, and a second $V_p = −2$ V is applied. This time, a higher $R_{SO}$ is measured followed by an abrupt decrease (Fig. 4c), corresponding to the higher branch of the full magnetization in Fig. 3f and to a 90° or 180° reversal of $M_{CoFe}$. This behavior was observed in eight out of nine switching attempts. All possible magnetization switching paths are illustrated in Fig. 4d, e, after positive and negative voltage pulses, respectively. In addition, the full switching data and statistics can be found in Supplementary Information Notes 5 and 6. All-in-all, these results demonstrate that the magnetization can be reversed and

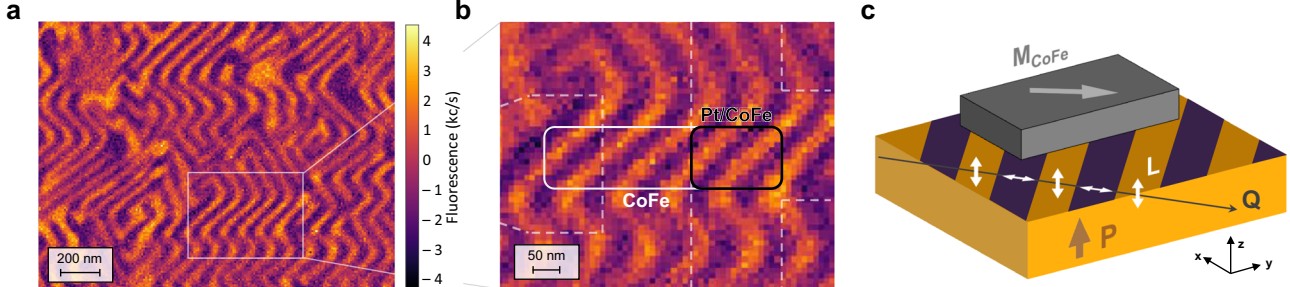

**Fig. 5 | Magnetic textures and spin cycloid in BiFeO₃. a** N-V magnetometry images on the bare BiFeO₃ surface where the MESO nanodevice was fabricated. **b** Zoomed region with a superimposed sketch (to scale) of the MESO nanodevice, revealing the possible complex magnetic behavior underneath the CoFe element. **c** Suggested coupling mechanism between $M_{CoFe}$ and the propagation direction of the cycloid Q.

read through voltage inputs and outputs, for both positive and negative $V_p$, fulfilling the initial MESO proposal. This writing/reading functionality is further explored in Supplementary Information Note 7, where $R_{SO}$ is consecutively probed for alternating $V_p$, in a steady state, with and without a static external magnetic field applied. The uncertainty imposed by possible diagonal magnetization orientations after switching, that give rise to a mix of SHE and PHE, contributes to the difficulty in associating one single $R_{SO}$ output value to a particular magnetization direction. In addition, the presence of a $R_{SO}$ baseline shift, described before, accounts for ~80 mΩ of the signal change, overlapping with the $R_{SO}$ difference between opposite magnetization orientations, expected to be in the order of 5–10 mΩ. Considering the baseline shift an inherent feature of the magnetic material when in direct contact with a switchable BiFeO₃, it is expected that future optimization of the output signal with materials beyond Pt, as discussed further ahead, should, in principle, be sufficient to surpass the magnitude of the baseline shift.

We further investigate the nature of the coupling between CoFe and BiFeO₃, which is expected to be responsible for the switchable $M_{CoFe}$. While the switching mechanism may be explained by coupling between $M_{CoFe}$ and the canted magnetization in BiFeO₃ (Ref. 13), the spin cycloid, reported in BiFeO₃ thin films grown on DyScO₃ substrates[26,27], may complicate this interpretation. Indeed, through scanning N−V magnetometry, we show in Fig. 5a that the cycloid is also present in our 30-nm-thick BiFeO₃, with a rotating AF order propagating in-plane in diagonal directions with a period of about ~70 nm, and changing its propagation direction (Q) by 90° in neighboring FE domains[28]. Here, the periodic variation of the magnetic stray field comes from the spin-density wave that is locked to the cycloid and perpendicular to the cycloidal plane defined by Q and P (Refs. 27,29). As exemplified in Fig. 5b, given the dimensions of the CoFe nanostructured element in our MESO devices, illustrated by the white rectangle, five full rotations of the AF order within each one of the single FE domain stripes are expected to interact with $M_{CoFe}$, with two of these rotations within the Pt/CoFe junction area (black rectangle). Within this area, the canted magnetization in BiFeO₃ should, in principle, average to zero. Since the magnetization in the CoFe is shown to be pulled in diagonal directions in the absence of external fields, as seen from the MFM and the electrical read-out characterization, we infer that $M_{CoFe}$ may in fact couple with Q (Fig. 5c). Through this type of coupling, a rotation of Q by ~90°/180°, for a partially or fully switched BiFeO₃, respectively, may be responsible for the reversal/rotation of $M_{CoFe}$.

## Discussion

We finish by discussing the reproducibility and non-deterministic aspects of our results, in light of the complex ferroelectric and magnetic textures of BiFeO₃. For two identical devices fabricated over different regions of the BiFeO₃, $M_{CoFe}$ may interact with completely different magnetic textures. Depending on this interaction, $M_{CoFe}$ may initially be pulled in different directions, so that the same voltage pulse polarity will drive different rotation/reversal paths, as observed in the PFM/MFM data in Supplementary Information Notes 2. This lack of correspondence between positive and negative voltage pulses with specific directions of $M_{CoFe}$ makes device-to-device reproducibility a real challenge. Not only that, but due to "maze-like" magnetic regions observed in Fig. 5a, some devices may not even exhibit a clear coupling with the BiFeO₃, a scenario further discussed in Supplementary Information Note 9. These issues may potentially be solved by better control of the FE domain structure of BiFeO₃ itself (or another multiferroic), ideally culminating in controlled single macroscopic domain regions with a coherent cycloid propagation[30,31] that can be effectively switched[32]. Alternatively, the overall absence of the cycloid could simplify the coupling mechanism, where $M_{CoFe}$ would couple with a uniform AF order in the multiferroic. Once these obstacles are surpassed, a systematic study would be desirable, using N-V magnetometry to simultaneously probe the magnetic texture in BiFeO₃ and the orientation of $M_{CoFe}$, and matching the switchable cycloid[32] with the magnetization state of the magnetic element.

Besides these fundamental issues, the future implementation of MESO logic will require additional improvements on both the ME and SO modules. Unlike STT or SOT current-based solutions, the reliability of the writing on MESO devices does not improve with larger input signals. In fact, as long as the BiFeO₃ can be engineered to switch robustly at lower voltages, the writing energies can be progressively reduced without compromising the reliability of the writing. The key elements to consider are the coupling between the magnet and the BiFeO₃, together with a soft magnet that can easily "follow" the magnetic motion in the multiferroic[15], while maintaining an overall thermally stable magnetization state and FE domain structure. Further miniaturization of the magnetic and SO elements to sub-100-nm features, together with the reduction of BiFeO₃ thickness, switching voltages (through La doping[33,34]), leakage currents (through Mn doping[35]) and switching pulse duration (down to tens of ns[36]) are pathways to reduce the switching energies to fJ and aJ ranges. Extended discussion on pulse duration implications and reproducibility of the coupling between CoFe and BiFeO₃ are presented in Supplementary Information Notes 8 and 9. In terms of endurance, the bottleneck comes from the degradation of the CoFe/BiFeO₃ interface with voltage cycling, given the possible formation of an oxidized or intermixed interfacial magnetic layer, as well as the degradation of BiFeO₃ itself. Solutions such as all-oxide epitaxial structures are one possible avenue to improve this[37]. On the SO module side, SCC output voltages between opposite magnetization states need to be at least comparable with the switching voltages of BiFeO₃, and ideally based on the ISHE instead of PHE, to make it scalable[19]. As initially discussed in Ref. 17, MESO devices interconnected as cascaded logic gates will require the output voltage of

one device to match the input switching voltage of the next one. For an input reading current of $100\,\mu A$, our SCC devices only show $\Delta V_{out} = 1\,\mu V$, while optimized Pt- and Ta-based devices reaching $30\,\mu V$ and $0.35\,mV$, respectively, have been reported[19,38]. Besides the output voltage, an increase in the corresponding output current, resulting from more efficient SCC phenomena, will also play an important role in faster switching. Nevertheless, additional efforts are required to reach hundreds of mV, potentially through all-electrical SCC in more exotic systems, such as topological insulators[39] and oxide heterostructures.

In conclusion, we have shown voltage-based writing and reading of magnetic states in a CoFe nanostructured element coupled with multiferroic $BiFeO_3$, representing the proof-of-principle for the MESO logic concept. Through a combination of PFM and MFM, we observe that the magnetization of CoFe can undergo 90° and 180° rotation/reversal, when the out-of-plane FE polarization of $BiFeO_3$ is switched using voltage pulses of $\pm 2\,V$. Using CoFe and Pt-based T-shaped nanostructures, we electrically detected the magnetization rotation/reversal, which leads to different voltage output states depending on the direction of CoFe magnetization. The presence of a spin cycloid with a period smaller than the size of the nanostructured magnet suggests that the magnetization control is driven by coupling with the propagating AF cycloid in $BiFeO_3$. While further work is required in terms of controllability and reproducibility of the switching, specifically regarding the ferroelectric and magnetic textures in $BiFeO_3$, these results provide a key step forward toward voltage-control of magnetization in nanoscale magnets, essential for future low-power spin-based logic and memory devices.

## Methods

### Sample preparation
The $DyScO_3(110)$ substrates were purchased from MTI Corporation and were cleaned with 20 min sonication at room temperature in acetone. The $DySsO_3$ substrates were bonded onto an Inconel carrier using silver paint. The silver paint was cured on a hot plate heated to 185 °C. The $SrRuO_3$, $La_{0.3}Sr_{0.7}MnO_3$, and $BiFeO_3$ were deposited using pulsed laser deposition with a laser fluence of approximately $1.5\,J/cm^2$ at 10 Hz and oxygen pressure of 150 mTorr. The $SrRuO_3$ was deposited at 690 °C and the $La_{0.3}Sr_{0.7}MnO_3$ and $BiFeO_3$ were deposited at 650 °C to minimize the Mn diffusion. The Co and Pt were deposited by physical vapor deposition in an in-situ magnetic field of ~300 to 400 Oe. A short vacuum break after the pulsed laser deposition (less than 45 s) was used to place the $DyScO_3$ Inconel carrier onto the physical vapor deposition sample holder configured with permanent magnets.

### Nanodevice fabrication
The devices were fabricated on Pt(10 nm)/CoFe(2.5 nm)/$BiFeO_3$(30 nm)/$La_{0.7}Sr_{0.3}MnO_3$(4 nm)/$SrRuO_3$(10 nm)/$DyScO_3$(110) samples (described above) with a multiple-step e-beam lithography, metal and oxide sputtering deposition, Ar-ion milling and lift-off process. Milling of the initial CoFe/Pt is performed with the ion gun at 10° with respect to the sample surface normal, an Ar flow of 15 s.c.c.m., an acceleration voltage of 50 V, a beam current of 50 mA and a beam voltage of 300 V. Side wall milling of nanostructures after lift-off is performed in the same conditions, with an angle of the ion gun at 80°. Control of the milling rates is achieved through real time end-point mass spectrometer and resistivities of milled films. Pt T-shaped nanostructures are fabricated using a positive PMMA 950A2 e-beam resist and deposited by magnetron sputtering with a rate of $1.25\,\text{Å}\,s^{-1}$, 80 W of power, $1.0 \times 10^{-8}$ mtorr of base pressure, 3 mtorr of Ar pressure. Isolation layer for wire bonder contact pads of $Al_2O_3$ is fabricated with a double-layer PMMA 495A4 + PMMA 950A2 resists and deposited with RF magnetron sputtering with a rate of $0.2\,\text{Å}\,s^{-1}$, 300 W of power, $1.0 \times 10^{-8}$ mtorr of base pressure, 3 mtorr of Ar pressure. All lift-offs were performed using acetone.

### Transmission electron microscopy and EDX
STEM and EDX studies were performed on Titan 60-300 Electron Microscope (FEI, Netherlands) at 300 kV accelerating voltage. The microscope was equipped by x-FEG, gun monochromator, retractable RTEM EDX detector (EDAX, USA) and HAADF detector. STEM images were acquired at nominal spot size 9, 10 mrad convergence angle, and −50 V relative gun lens potential. EDX mapping was done at a nominal spot size of 6 and −15 V gun lens potential to provide a sufficient count rate. The cross-sections of the devices were prepared by a standard FIB lamellae fabrication technique: a protective Pt layer was deposited first by e-beam followed by ion-beam deposition, lamellae of ~2 μm thickness were undercut and transferred onto a copper half-grid, thinned there to ~200 nm by 30 keV $Ga^+$ beam, and finally polished to ~20 nm at 5 keV.

### Piezoresponse force microscopy
PFM experiments were conducted with an atomic force microscope (Nanoscope V multimode, Bruker). Two external lock-in detectors (SR830, Stanford Research) were used to simultaneously acquire vertical and lateral piezoresponses. An external source (DS360, Stanford Research) was used to excite the $La_{0.7}Sr_{0.3}MnO_3/SrRuO_3$ bottom electrode (ac 0.6 V peak-to-peak at 35 kHz) while the conducting Pt-coated tip was grounded. Pt-coated tips (Budget Sensors) with $40\,Nm^{-1}$ cantilevers were chosen for these images. For the experiments on the bare $BiFeO_3$ surface (Fig. 2a), the same source was used to write domains with a dc voltage while scanning. For the experiments on devices (Fig. 2c), write voltage pulses (1 s) were applied while the tip was in contact with Pt/CoFe top electrode but not scanning.

### Magnetic force microscopy
The MFM observation of the Pt/CoFe nanostructures was performed in a setup under low pressure, of the order of $P = 10^{-6}$ mbar. Images were obtained at room temperature using magnetic tips in a double-pass tapping–lift mode, detecting the phase shift of the second pass after a topographic measurement and thus probing the magnetic field gradient along the vertical direction. Tips were fabricated in our laboratory by depositing a magnetic coating on commercial silicon tips with magnetic sputtering, whose thicknesses were in the range 3–23 nm for CoFeB, which we selected for their particularly low degree of perturbation on the magnetic configurations under observation and improved signal-to-noise ratio, with quality factor $Q = 1500$ and spring constant $k = 0.4\,N\,m^{-1}$.

### Electrical characterization
Transport measurements were performed in a Physical Property Measurement System from Quantum Design, using a "d.c. reversal" technique with a Keithley 2182 nanovoltmeter and a 6221 current source at 300 K. The input current $I_{in}$ for the measurements was 20 μA. Gate voltage pulses were applied with a Keithley 2636B Sourcemeter, with a pulse duration of 200 μs. Samples were mounted in a rotatable sample stage and the external magnetic field $B_{ext}$ is applied with a superconducting solenoid magnet. Devices were contacted using a wire bonder, with Au wire heated at 70 °C and a wedge bonding force of 20 cN.

### Scanning N−V magnetometry
The antiferromagnetic spin textures of $BiFeO_3$ were imaged using a commercial scanning N−V magnetometer (ProteusQ™, Qnami AG) operated under ambient conditions. In our setup, the scanning tip is a commercial all-diamond probe with a single N−V defect at its apex integrated on a quartz tuning fork (Quantilever™ MX, Qnami AG). The diamond tip is integrated into a tuning-fork-based atomic force microscope combined with a confocal microscope optimized for single N−V defect spectroscopy.

## Data availability

All data are available in the main text and the Supplementary Information. Additional data related to the findings in this study can be requested from the authors.

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

## Acknowledgements

We acknowledge C. Rufo, R. Llopis, and R. Gay for technical assistance with the sample fabrication and electrical characterization. This work is supported by Intel Corporation through the Semiconductor Research Corporation under MSR-INTEL TASK 2017-IN-2744 and the 'FEINMAN' Intel Science Technology Center, by the Spanish MCIN/AEI/10.13039/501100011033 and by ERDF A way of making Europe (Project No. PID2021-122511OB-I00 and Maria de Maeztu Units of Excellence Program No. CEX2020-001038-M). D.C. Vaz acknowledges support from the European Commission for a Marie Sklodowska-Curie individual fellowship (Grant No. 892983-SPECTER). W.Y. Choi acknowledges postdoctoral fellowship support from 'Juan de la Cierva Formación' program by the Spanish MCIN/AEI (Grant No. FJC2018-038580-I). The team at Laboratoire Albert Fert acknowledges the support from the French Agence Nationale de la Recherche (ANR) through the project TATOO (ANR-21-CE09-0033), the European Union's Horizon 2020 research and innovation program under the Grant Agreements No. 964931 (TSAR), a public grant overseen by the ANR as part of the "Investissements d'Avenir" program (Labex NanoSaclay, reference: ANR-10-LABX-0035), and the Sesame Ile de France IMAGeSPIN project (No. EX039175). The

team at the Department of Physics in University of California, Berkeley, acknowledges the support from the SRC JUMP program.

## Author contributions

F.C. and I.A.Y. proposed and supervised the study with the help of L.E.H., D.C.V., and C.-C.L. J.J.P. performed the growth and deposition of materials, with inputs from Y.-L.H., B.P., and R.R. D.C.V., W.Y.C., I.G., and I.C.A. designed and fabricated the nanodevices, with inputs from B.P., C.-C.L., D.E.N., H.L., P.D., S.B.C., T.A.G., I.A.Y., and F.C. A.C. performed the EDX and TEM analysis. A.V., K.B., S.F., and V.G. performed the PFM, MFM, and NV magnetometry experiments and analyzed the data with inputs from M.B. D.C.V. performed the electrical characterization and analyzed the data with the help of all authors. D.C.V. wrote the paper with inputs from all authors. All authors discussed the results and contributed to their interpretation.

## Competing interests

The authors declare no competing interests.
