## [Peer Review File · Nature Communications]

Voltage-based magnetization switching and reading in magnetoelectric spin-orbit nanodevicesEditorial Note: This manuscript has been previously reviewed at another journal that is not operating a transparent peer review scheme. This document only contains reviewer comments and rebuttal letters for versions considered at *Nature Communications*.

REVIEWER COMMENTS

Reviewer #1 (Remarks to the Author):

The authors tried to address my concerns by using reasonable arguments. While it would be better to have more experimental results, I agree that the current form is publishable for Nature Communications.

Reviewer #3 (Remarks to the Author):

The authors have made a careful response to the reviewers' comments, while did very little in revising their manuscript. There are still some obvious defects in the present manuscript.

For instance, the authors boasted in abstract that "Here, we demonstrate voltage-based magnetization switching and reading in nanodevices at room temperature, enabled by exchange coupling between multiferroic BiFeO₃ and ferromagnetic CoFe, for the writing, and spin-to-charge current conversion between CoFe and Pt, for the reading." While the both writing and reading processes are not successful. Specially, the reading process can only obtained based on R_{so}-H curves (show H_c, H_b switching), which is definably not enough for demonstrating a reading operation. I would suggest the authors mentioned this point in abstract, as well as modify the manuscript and tile accordingly, otherwise the manuscript could be rather misleading.

Since the authors failed to demonstrate the pure ΔR_{so} switching induced by input voltage, due to the occurrent of large baseline shift, the authors should give an in-depth discussion on the mechanism and how to suppress the baseline shift signal. For demonstrating pure ΔR_{so} , I would suggest the authors at least show reversible ΔR_{so} switching induced by magnetic field pulse only for several repeating cycles, wherein the baseline shift can be excluded.

REPLY TO REVIEWERS (13-11-2023)

Reviewer #1 (Remarks to the Author):

The authors tried to address my concerns by using reasonable arguments. While it would be better to have more experimental results, I agree that the current form is publishable for Nature Communications.

We thank the reviewer for his/her assessment and for supporting the publication of our manuscript in its current form.

Reviewer #3 (Remarks to the Author):

The authors have made a careful response to the reviewers' comments, while did very little in revising their manuscript. There are still some obvious defects in the present manuscript.

For instance, the authors boasted in abstract that "Here, we demonstrate voltage-based magnetization switching and reading in nanodevices at room temperature, enabled by exchange coupling between multiferroic BiFeO₃ and ferromagnetic CoFe, for the writing, and spin-to-charge current conversion between CoFe and Pt, for the reading." While the both writing and reading processes are not successful. Specially, the reading process can only obtained based on R_{so}-H curves (show H_c, H_b switching), which is definably not enough for demonstrating a reading operation. I would suggest the authors mentioned this point in abstract, as well as modify the manuscript and tile accordingly, otherwise the manuscript could be rather misleading.

Since the authors failed to demonstrate the pure ΔR_{so} switching induced by input voltage, due to the occurrent of large baseline shift, the authors should give an in-depth discussion on the mechanism and how to suppress the baseline shift signal. For demonstrating pure ΔR_{so} , I would suggest the authors at least show reversible ΔR_{so} switching induced by magnetic field pulse only for several repeating cycles, wherein the baseline shift can be excluded.

We thank the reviewer for his/her assessment of the manuscript. Following the comments made regarding the "steady state" reading feature of our devices, we added a line in the abstract mentioning that reading is achieved through magnetic field sweeps. Additionally, we have added the following paragraph to the main text elaborating on the baseline shift role in the reading process:

"The uncertainty imposed by possible diagonal magnetization orientations after switching, that give rise to a mix of SHE and PHE, contributes to the difficulty in associating one single R_{so} output to a particular magnetization direction. In addition, the presence of a R_{so} baseline shift, described before, accounts for ~80 mΩ of the signal change, overlapping with the R_{so} difference between opposite magnetization orientations, expected to be in the order of 5 to 10 mΩ. Considering the baseline shift an inherent feature of the magnetic material when in direct contact with a switchable BiFeO₃, it is expected that future optimization of the output signal with materials beyond Pt, as discussed further ahead, should, in principle, be sufficient to surpass the magnitude of the baseline shift."

Further clarification is included in Supplementary Note 7.

We emphasize that, according to our assessment, the baseline shift appears to be an inherent property arising from the resistance change of CoFe when in contact with a switchable BiFeO₃, as described in the main text:

“The shift in baseline resistance can be explained by slight modulation of the resistivity of CoFe, either due to a static field effect from the remanent polarization in the BiFeO₃ (Ref.²¹), or strain induced by different ferroelastic domains.”

Further investigation on this feature is being carried out to test out this possibility. Although we postulate that this feature may be unavoidable in magnetic/multiferroic systems, further optimization using alternative SO materials, such as Ta, complex oxides, or topological insulators, should contribute to an increase of the magnitude of the output signals of MESO devices, so that, in future devices, ΔR_{SO} should be much larger than the baseline shift contribution.

For added clarity and as requested by the reviewer, we include Figure S9b in the supplementary information (also shown below), where the baseline shift (~ 80 m Ω) is subtracted from the raw ΔR_{SO} data. One can observe that, at times, a finite ΔR_{SO} is observed after $V_p = -2$ V (red circles), implying changes in the magnetization direction. However, without the insights provided by the magnetic field sweeps, this data, by itself, is insufficient to claim magnetization switching.

Figure S9b - ΔR_{SO} as a function of alternating $V_p = \pm 2$ V in the absence of an external magnetic field, subtracted by the baseline shift of 80 m Ω .

REVIEWER COMMENTS

Reviewer #3 (Remarks to the Author):

The authors have made a detailed response to the reviewers queries, while they should made more revisions on the abstract, indicating the shortcomings in this work. For instance, the electric driven magnetization switching effect is not very controllable, and the reading signals are too small. The authors should also provide a ΔR_{so} switching curve that was induced by pure magnetic pulse, and compare it with the electric induce ΔR_{so} curve after subtracting the based line shift (shown in the response letter). This can verify that whether the ΔR_{so} (shown in response letter) is really from the spin-orbit coupling signals or not. The manuscript can be accepted for publication after carefully addressing these two issues.

REPLY TO REVIEWERS (14-01-2024)

Reviewer #3 (Remarks to the Author):

The authors have made a detailed response to the reviewers queries, while they should made more revisions on the abstract, indicating the shortcomings in this work. For instance, the electric driven magnetization switching effect is not very controllable, and the reading signals are too small.

We thank the reviewer for his/her additional comments. A new sentence has been added to the abstract highlighting the non-deterministic nature of the switching and the fact that the output signal is small, as requested by the reviewer.

The authors should also provide a ΔR_{so} switching curve that was induced by pure magnetic pulse, and compare it with the electric induce ΔR_{so} curve after subtracting the based line shift (shown in the response letter). This can verify that whether the ΔR_{so} (shown in response letter) is really from the spin-orbit coupling signals or not. The manuscript can be accepted for publication after carefully addressing these two issues.

An extensive analysis of the R_{so} signal vs. external magnetic field is presented in Figure 3 (main text) and Figure S6 (Supplementary Information), in particular, where R_{so} values are mapped to the magnetization directions of the CoFe element (see Figure 3c). Here, we can see the change in output signal (R_{so}) for known magnetization directions.

However, as noted in previous replies, the ΔR_{so} vs. voltage pulse, presented in Fig. S9 (Supplementary) cannot, by itself, represent a solid proof of magnetization switching, since absolute values of ΔR_{so} alone can be associated with different magnetization directions. In this Figure (S9a), considering the ΔR_{so} values obtained after applying a voltage pulse, the respective possible magnetization directions are represented in blue and red arrows, highlighting that this measurement by itself is not sufficient to prove magnetization switching, but serves only as an indicative experiment of what can be obtained within the current limitations of the devices shown.

In the present manuscript, we do not claim the existence of deterministic switching or a perfectly optimized device. We have observed magnetization switching solely induced by a voltage pulse, and in the same device we are able to obtain different voltage outputs after switching. We believe that the shortcomings mentioned by the reviewer, and by ourselves, have been adequately presented and explained throughout the manuscript, with several suggestions on how to improve them. In our view, this combination is, as it is, an important step that opens new ground for more experiments and articles that can build upon our work.